# Functions and Regulation of Meiotic HORMA-Domain Proteins

**DOI:** 10.3390/genes13050777

**Published:** 2022-04-27

**Authors:** Josh P. Prince, Enrique Martinez-Perez

**Affiliations:** 1Meiosis Group, MRC London Institute of Medical Sciences, London W12 0NN, UK; j.prince@lms.mrc.ac.uk; 2Faculty of Medicine, Imperial College London, London W12 0NN, UK

**Keywords:** meiosis, HORMA domain, axial element, meiotic chromosomes, meiotic checkpoints

## Abstract

During meiosis, homologous chromosomes must recognize, pair, and recombine with one another to ensure the formation of inter-homologue crossover events, which, together with sister chromatid cohesion, promote correct chromosome orientation on the first meiotic spindle. Crossover formation requires the assembly of axial elements, proteinaceous structures that assemble along the length of each chromosome during early meiosis, as well as checkpoint mechanisms that control meiotic progression by monitoring pairing and recombination intermediates. A conserved family of proteins defined by the presence of a HORMA (HOp1, Rev7, MAd2) domain, referred to as HORMADs, associate with axial elements to control key events of meiotic prophase. The highly conserved HORMA domain comprises a flexible safety belt sequence, enabling it to adopt at least two of the following protein conformations: one closed, where the safety belt encircles a small peptide motif present within an interacting protein, causing its topological entrapment, and the other open, where the safety belt is reorganized and no interactor is trapped. Although functional studies in multiple organisms have revealed that HORMADs are crucial regulators of meiosis, the mechanisms by which HORMADs implement key meiotic events remain poorly understood. In this review, we summarize protein complexes formed by HORMADs, discuss their roles during meiosis in different organisms, draw comparisons to better characterize non-meiotic HORMADs (MAD2 and REV7), and highlight possible areas for future research.

## 1. Introduction

Proteinaceous structures, called axial elements, are a defining structural feature of meiotic chromosomes [1]. The assembly of these structures begins with the loading of meiosis-specific versions of the cohesin complex at meiotic onset, providing a scaffold for the recruitment of additional meiosis-specific proteins, including HORMADs, whose incorporation into axial elements is essential for subsequent meiotic events. This includes initiating meiotic recombination and formation of the synaptonemal complex (SC), a tripartite structure that assembles at the interface between aligned homologue pairs and is required for crossover formation [2]. *Saccharomyces cerevisiae* Hop1 was the first meiotic HORMAD to be identified, with mutants lacking Hop1 displaying a reduction in crossover events and impaired SC formation [3]. Subsequent studies demonstrated that Hop1 is required for the formation of DNA double strand breaks (DSBs) that initiate meiotic recombination, to promote their repair using the homologous chromosome as a repair template, and for checkpoint control of meiotic prophase [4,5,6]. Similar roles have now also been described for Hop1 orthologues in *Saccharomyces pombe* (Hop1) [7], mice (HORMAD1 and HORMAD2) [8,9,10,11], plants (ASY1) [12], and nematodes (HIM-3, HTP-1, HTP-2 and HTP-3) [13,14,15,16] (Table 1). In addition to their roles in pairing, recombination and meiotic checkpoints, *Caenorhabditis elegans* HORMADs are also required for ensuring normal levels of cohesin on axial elements [17], for the acquisition of axis-associated histone marks [18], and for the two-step release of sister chromatid cohesion during the meiotic divisions [17,19,20]. Thus, HORMADs control critical chromosomal events of the meiotic program, but despite their fundamental importance for fertility, knowledge of the mechanisms by which HORMADs control meiotic events remains limited.

The biochemical and structural features of proteins containing a HORMA domain are widely conserved. Therefore, we start this review by describing how conformational changes in Mad2 and Rev7, the best characterized HORMA-domain proteins, control the formation and disassembly of protein complexes involved in chromosome segregation and DNA repair in somatic cells. We next describe the protein complexes formed by meiotic HORMADs in different organisms, focusing on their roles in the assembly of axial elements, meiotic recombination, synaptonemal complex formation, checkpoint regulation of meiotic progression, and the role of posttranslational modifications in regulating meiotic HORMADs’ functions. Table 2 offers a detailed summary of the interactors for Mad2, Rev7 and meiotic HORMADs. Throughout the review, we highlight gaps in our current understanding of the mechanisms by which HORMADs control multiple aspects of meiotic chromosome behavior.

## 2. The HORMA Domain: A Protein-Protein Interaction Module for Regulatory Mechanisms

The HORMA domain was first identified and named on the basis of a primary sequence similarity between yeast Hop1, Rev7, and Mad2 [21]. Mad2 is a key component of the spindle assembly checkpoint that regulates chromosome segregation during mitosis and meiosis [22], while Rev7 is a multifunctional protein involved in different DNA repair pathways, including the translesion synthesis and shielding complexes [23]. Mad2 and Rev7 are short proteins, consisting exclusively of their HORMA domains, which can be further divided into the following two distinct domains: a core consisting of three central α-helices (αA-C) and a three-stranded β-sheet (β4-6), and a flexible C-terminal domain called the safety belt (Figure 1). In solution, recombinant monomeric Mad2 preferentially adopts an open Mad2 (O-Mad2) conformation, where β8 of the safety belt domain interacts with β6 [24,25]. The addition of a short peptide motif, termed closure motif (CM), from one of its binding partners leads to a significant conformational rearrangement, promoting the safety belt (β8′ and β8″) to interact with β5 on the opposite side of the protein. This results in a closed Mad2 (C-Mad2) conformation, in which the CM is topologically entrapped by the safety belt [26]. This rearrangement has been subsequently identified in Rev7 and Hop1, and is believed to be a common property of all HORMA domain-containing proteins [27,28,29].

Reversing the stable interaction between C-Mad2 and a CM requires an energy dependent unfolding of the HORMA domain. The disassembly of the Mad2-containing mitotic checkpoint complex (MCC), which is required for spindle assembly checkpoint inactivation, is catalyzed by the conserved protein Pch2 or its mammalian orthologue TRIP13, members of the AAA + ATPase superfamily. Through the hydrolysis of ATP, Pch2/TRIP13 facilitates the opening of C-MAD2, in a process that also requires the adaptor protein p31^comet^ [31]. Recent cryo-EM structures of the TRIP13-MAD2-CDC20-p31^comet^ complex suggested this would occur through an interaction of MAD2’s N-terminus with TRIP13, leading to the destabilization of C-MAD2 [32]. The deletion of these disordered N-terminal residues in MAD2 leads to a preference for the closed conformation and spindle assembly checkpoint activation defects in cells, verifying the importance of these residues in its conformational rearrangement [33]. Similar evidence suggests that TRIP13 and p31^comet^ also function to open C-REV7 [34,35], and to modify the folding of meiotic HORMADs (see below), thus, pointing to a universally conserved mechanism for HORMAD opening.

In addition to the canonical CM binding mediated through the HORMADs safety belt domain and the N-terminal interaction with TRIP13/Pch2, additional surfaces on MAD2 and REV7 have been shown to mediate protein-protein interactions (Figure 1, Table 2). Both MAD2 and REV7 can undergo homo- and heterodimerization, mediated around the αC helix [25,36] (Table 2). In Mad2, the mutagenesis of this dimerization interface results in a loss of function, underlying the importance of these interactions [37]. Human REV7 has also been shown to interact with REV1 through residues on β5 and the safety belt (β8′ and β8″), and SHLD2 through β6 (Figure 1) [38,39]. Interestingly, these interactions are conformationally specific. By coupling HORMA domain conformation to the presence of specific intermediates of chromosome metabolism, cells can rapidly react to different situations, by triggering the assembly or disassembly of HORMA-containing protein complexes. This property has made HORMADs key components of different signaling mechanisms [29]. 

## 3. Protein Complexes Formed by Meiotic HORMADs

In contrast with Mad2 and Rev7, which consist exclusively of a HORMA domain, meiotic HORMADs contain additional N- and C-terminal regions. These domains have the potential to both affect the interactions mediated by the HORMA domain, and to act as platforms for recruiting interactors beyond those bound by the safety belt mechanism or other surfaces of the HORMA domain. Considering the large number of proteins reported to interact with human REV7 (Table 2), it seems likely that the current low number of confirmed meiotic-HORMAD interactors is an underrepresentation of the actual number, especially given the multiple roles HORMADs play during meiosis. 

A conserved feature of the C-terminal region of meiotic HORMADs is the presence of CMs that promote the formation of HORMAD oligomers on chromosome axes [27,40]. These CMs are also proposed to induce the formation of a self-closed conformation (SC-HORMAD), in which the HORMA domain binds the CM on its own C-terminus [27,41] (Figure 2 and Figure 3). We note that, although interactions between the HORMA domain and their internal CMs have been identified, currently there is no direct evidence showing that this occurs via the safety belt-CM mechanism in cis (i.e., structure of a full length HORMAD binding its internal CM). Similarly, structural evidence for meiotic HORMADs displaying a stable open conformation, as observed for Mad2, is currently lacking. However, the HORMA domain of Hop1 adopts two distinct monomeric conformations in solution: one consistent with a closed conformation and a second in which the safety belt is disengaged from the HORMA domain [27]. This conformation, referred to as unbuckled, is thought to represent an intermediate that allows the binding of the safety belt to a CM. Below, we summarize the studies that are starting to elucidate how meiotic HORMADs and their interactors control different chromosomal events of meiosis.

## 4. Binding of Meiotic HORMADs to Axial Elements

In early meiotic prophase, HORMADs and other meiosis-specific proteins localize to the chromosomal axis in a cohesin-dependent manner. Early studies in yeast demonstrated that the axis component Red1 is required for recruiting Hop1 to axial elements [42] and recently, a CM present on Red1 was shown to mediate Hop1 binding [27] (Figure 2A). Red1 orthologues in *S. pombe* (Rec10) [27,43], mammals (SYCP2), and plants (ASY3) also contain CMs that directly interact with meiotic HORMADs (Figure 2B,D and Table 2) [44]. In mammals, in vitro experiments show that the HORMA domain of HORMAD1 interacts with the CM on the C-terminus of HORMAD1 and HORMAD2 [40], and that the HORMA domain of HORMAD2 interacts with a CM on its own C-terminus and to one present in SYCP2 [44] (Figure 2B). In vivo, HORMAD2 recruitment is largely dependent on HORMAD1 [10], suggesting that HORMAD2 may also bind the CM on HORMAD1’s C-terminus. HORMAD1 is recruited to axial elements in the absence of SYCP2, but not of meiotic cohesin [45], suggesting that cohesin directly interacts with HORMAD1. The colocalization of meiotic HORMADs and cohesin on spread nuclei imaged by super-resolution microscopy, as well as immunoprecipitation experiments, provide further support for an interaction between cohesin and meiotic HORMADs in worms and mammals [40,45,46]. In *C. elegans*, which lack a Red1 homologue, cohesin is required to recruit HTP-3 to axial elements [14] and CMs on the C-terminus of HTP-3, in turn, recruit HORMADs HIM-3, HTP-1, and HTP-2 to the axis [40] (Figure 2C). In addition, HTP-1 and HTP-2 are also recruited to a CM on the C-terminus of HIM-3. Structural studies of the HORMA domain of HIM-3, HTP-1, and HTP-2, bound to CMs from HTP-3, confirm binding by the canonical safety belt mechanism around the CMs peptides [40]. Therefore, the main mechanism of recruiting meiotic HORMADs to axial elements is conserved, involving a closed conformation of the HORMA domain around a CM present on a chromosome-bound protein. However, the exact mechanism by which HORMADs, or HORMADs bound to Red1/SYCP2/ASY3, interact with cohesin has only been reported for *S. pombe*, where Hop1 binds a CM on Rec10(Red1) [43] and Rec10 binds to the N-terminus of Rec11 (a meiosis-specific version of the cohesin protein Scc3) [47].

Hop1 contains a zinc finger motif on its C-terminus, displays DNA binding activity in vitro [48], and ChIP analysis shows that Hop1 can be recruited to chromatin in complex with Red1 in the absence of cohesin [49]. This cohesin-independent recruitment of Hop1 is mediated by a predicted PHD domain on Hop1’s C-terminus that encompasses the previously identified zinc finger motif [50]. A cohesin independent pathway of HORMAD recruitment to chromosomes may also operate in plants, as loading of ASY1 is reduced, but not eliminated, in mutants lacking meiotic cohesin [51]. Therefore, meiotic chromosomes may contain pools of HORMADs recruited by cohesin-dependent and -independent mechanisms, as well as recruited to CMs on different interactors (for example, Hop1 bound to CMs on Hop1 or Red1). Elucidating all the mechanisms by which HORMADs are recruited to axial elements, and how pools of HORMADs bound to CMs on different interactors correlate to specific functions remains an important goal for future studies.

## 5. Pch2/TRIP13 Regulates the Assembly and Disassembly of Protein Complexes Containing Meiotic HORMADs

Mutants lacking Pch2/TRIP13 in yeast, mammals, plants and nematodes display defects in multiple meiotic events, including altered localization of meiotic HORMADs and impaired checkpoint activity [52,53,54,55]. In yeast, mammals, and plants, meiotic HORMADs bind to axial elements during early meiotic prophase and are then largely removed coinciding with SC assembly. However, in the absence of PCH2/TRIP13, meiotic HORMADs persist on synapsed regions during the pachytene stage, suggesting that PCH2/TRIP13 promotes HORMAD removal from the axis [54,55] (Figure 3). 

Recent studies in *Arabidopsis* and rice are particularly informative on the effect of PCH2 in regulating the behavior of meiotic HORMADs. In rice mutants lacking PCH2, PAIR2 (Hop1) fails to load on axial elements [56], while in *Arabidopsis pch2* mutants, a pool of ASY1 remains in the cytoplasm and axis-associated ASY1 is not removed following SC installation [55,57]. The C-terminal region of ASY1, containing a CM, also acts to target the protein to the nucleus; therefore, the binding of ASY1’s HORMA domain to its own CM would result in a SC-ASY1 that would prevent nuclear import [41] (Figure 3). The unlocking of this cytoplasmic SC-ASY1 by PCH2 promotes nuclear import, while nuclear soluble PCH2 maintains a pool of unbuckled ASY1 that can be loaded to the axial elements. Later in prophase, PCH2 localizes to the SC, presumably unlocking ASY1 bound to the CM of ASY3, and therefore releasing ASY1 from axial elements. This model is further supported by the finding that p31^comet^, a PCH2 cofactor required for opening MAD2 to release it from the MCC (see above), is also required to promote the nuclear import of ASY1 in early prophase and for its chromosome removal during pachytene in *Arabidopsis* [58], and for pairing and recombination in rice [59]. 

In yeast, Pch2 interacts with Hop1 in vitro [60], and a cytoplasmic pool of Pch2 is proposed to ensure that unbuckled Hop1 is available to associate with Red1 and to incorporate into the axis [61]. Similar to the situation in plants, yeast Pch2 is proposed to ensure that soluble unbuckled-Hop1 is available for incorporation into the axis and to induce Hop1 removal from the axis following SC assembly [62]. This process is mediated by the SC component Zip1, which recruits Pch2 to chromosomes [63,64]. In worms, meiotic HORMADs persist bound to axial elements after SC assembly and their localization is not controlled by PCH-2 [65], suggesting that additional factors may be involved in regulating their conformational changes. Nonetheless, in the absence of PCH-2 or CMT-1 (p31^comet^), the fidelity of SC assembly, a process controlled by meiotic HORMADs (see below), is compromised [65,66], thus, opening the possibility that PCH-2 also regulates meiotic HORMADs in worms. Overall, the studies mentioned above are consistent with Pch2/TRIP13 acting as a master regulator of the timely assembly and disassembly of complexes containing meiotic HORMADs, by unfolding C-HORMADs bound to a CM to produce an unbuckled intermediate. However, so far, an unbuckled conformation has only been observed for a version of yeast Hop1 consisting exclusively of its HORMA domain [27]. Clarifying how Pch2 interacts with full length meiotic HORMADs to regulate their conformation and functions remains an important goal.

## 6. HORMADs Roles in Meiotic Recombination

Of the multiple functions exerted by HORMADs during meiosis, their role in promoting DSB formation during the early stages of meiotic recombination is perhaps the best understood in terms of protein complex formation. Meiotic DSB formation is initiated by Spo11 [67], a transesterase that is controlled by multiple cofactors, which regulate its localization to chromosomes, as well as its timing and levels of activity. Studies in yeast showed that the MMR (Mre2, Mei4, Rec114) complex is required for Spo11-dependent DSB formation, that the recruitment of Mei4 and Rec114 to axial elements depends on Mer2, and that Mer2 recruitment, in turn, depends on Hop1 [68]. Mer2 is evolutionary conserved [69] and its localization to chromosomes in mammals also depends on HORMAD1 [70], consistent with meiotic HORMADs acting as the platforms for recruiting the MMR complex to axial elements. Recent studies on *S. pombe* and *S. cerevisiae* have provided important insights into the interaction between Hop1 and Mer2. In *S. pombe*, sequences on the C-terminus of Hop1 are required for the interaction between Hop1 and Rec15 (Mer2), and Hop1 enhances the formation of a complex between Rec10 (Red1) and Rec15 (Mer2) to promote DSB formation [43]. Pull-down experiments with purified *S. cerevisiae* Hop1 and Mer2 demonstrated a weak interaction between the proteins, but the amount of Mer2 greatly increased when the pull-down was performed with a Red1-Hop1 complex instead of Hop1 alone, while no interaction was detected between Mer2 and Red1 alone [71]. An increase in Mer2 pull-down with Hop1 was also detected using Hop1 with a mutation in its CM that prevents the acquisition of the self-closed conformation. Finally, crosslinking experiments identified interactions between Mer2, and the C-terminus and HORMA domain of Hop1. These observations suggest that Mer2 binding to Hop1 occurs in the context of Hop1 bound to the Red1’s CM (i.e., associated with axial elements), and not with self-closed soluble Hop1 [71]. Thus, the regulation of the self-closed conformation can be used as an on–off switch to control the accessibility of domains that recruit interactors on the C-terminus, for example, by allowing the binding of an interactor only when the CM on the C-terminus is not bound to the HORMA domain of the same molecule (Figure 4). It is likely that this on–off switch could similarly regulate the accessibility of surfaces on the HORMA domain itself, as observed for REV7 [38]. 

Although no Mer2 homologue has so far been identified in *C. elegans*, meiotic HORMADs HTP-3 and HTP-1 have been proposed to promote DSB formation [14,15,16]. HTP-3 interacts with recombination proteins MRE-11 and RAD-50 [14] and promotes DSB formation, in a manner that is independent of its role in recruiting HTP-1, HTP-2, and HIM-3 to axial elements [40]. Therefore, HTP-3 could act as a platform for recruiting DSB-promoting factors, in a manner analogous to the way in which Hop1 recruits Mer2 in other organisms. The mechanisms by which HTP-1 could promote DSB formation are not understood, but the severe reduction in recombination intermediates observed in *htp-1* mutants could be related to the premature exit from DSB-competent stages in the absence of HTP-1 and to its potential role in preventing sister-mediated DSB repair [15,16], rather than a direct role of HTP-1 in recruiting DSB-promoting factors to the axis.

The ability of HORMADs to act as platforms that recruit DSB-promoting factors to axial elements can explain how DSB formation ceases once HORMADs are ejected from the axis. In yeast and mammals, DSB formation induces SC assembly, a process that coincides with HORMAD ejection from axial elements. Therefore, by promoting DSB formation and subsequent SC assembly, HORMADs set up in motion their eventual removal from chromosomes. In worms, SC assembly is independent of DSB formation [72] and HORMADs remain associated with the axis through pachytene. Nonetheless, changes in SC status, induced by recombination, are proposed to be sensed and/or transmitted by HTP-1 in a manner that controls the activity of the DSB-promoting CHK-2 kinase [73]. Thus, conceptually, the mechanism that regulates DSB formation competence in worms appears similar to that of yeast and mammals, in that it may involve SC-triggered changes in HORMAD behavior.

During meiotic recombination, HORMADs also play an important role downstream of DSB formation, by promoting the use of a homologous chromosome, instead of a sister chromatid, as a template for DSB repair. In yeast, the C-terminus of Hop1 recruits the Mek1 kinase to axial elements, promoting Mek1 dimerization and activation, which leads to the downregulation of the mitotic Rad51 recombinase to favor Dmc1-based DSB repair [5,74]. The recruitment of Mek1 to the C-terminus of Hop1 requires the phosphorylation of this region by the ATM/ATR kinases [75], evidencing how posttranslational modifications on HORMADs control protein recruitment to axial elements (see below). In *Arabidopsis*, ASY1 is not required for DSB formation, but is thought to promote DMC1-dependent DSB repair to ensure inter-homologue recombination [12] and also controls crossover distribution across the genome [76]. HORMAD1 and HORMAD2 in mammals [77,78,79,80] and HIM-3 and HTP-1 in worms [15,16,81] are also proposed to prevent inter-sister recombination, but the precise mechanisms are not understood.

## 7. HORMADs Involvement in SC Assembly 

In all the organisms studied, mutants lacking meiotic HORMADs showed reduced and/or improper (between non-homologous chromosomes) SC assembly. In yeast and mammals, where SC assembly depends on DSB formation, meiotic HORMADs mostly promote synapsis indirectly by ensuring DSB formation. However, HORMAD1 also promotes SC assembly independently of its role in DSB formation [9]. A requirement for HORMADs in promoting SC assembly in a DSB-independent manner is clearly established in *Arabidopsis*, where *asy1* mutants undergo DSB formation, but show impaired SC assembly [12], and in *C. elegans*, where SC assembly is independent of DSB formation and HIM-3 and HTP-3 are required for SC assembly [14,81]. The role of HTP-3 in this process is exerted indirectly, by recruiting HIM-3 to four CMs on its C-terminus [40]. In contrast to the role of HIM-3 in promoting SC assembly, HTP-1 limits this process until homology recognition, a process also promoted by HTP-1, is satisfied, thus, preventing non-homologous synapsis [15,16]. In addition, HTP-2 also appears to promote SC assembly, as synapsis is reduced and delayed in *htp-1 htp-2* double mutants compared to *htp-1* mutants [15,20]. Super-resolution microscopy shows that the HORMA domain of HIM-3 is situated in the interface between the axis and SC central region components, while HTP-3 and HTP-1/2 HORMAs localize in the proximity of cohesin [82]. Thus, worm HORMADs could act as a bridge between cohesin and SC components, in which HIM-3 would mediate the interaction with central region proteins. How HORMADs promote SC assembly in organisms where these proteins are largely removed from axial elements coinciding with the onset of synapsis is not known, but it could involve a “licensing SC assembly” step, for example, by promoting post-translational modifications on the axial components that are not removed during SC assembly.

## 8. Checkpoint Regulation of Meiotic Prophase by Meiotic HORMADs

HORMADs implement quality control of meiosis by monitoring meiotic chromosome metabolism intermediates and by transmitting signals that regulate the activity of kinases that orchestrate meiotic progression [83,84]. In yeast, Hop1 is phosphorylated by Tel1/Mec1 (ATM/ATR) in a DSB-dependent fashion, inducing recruitment and activation of the Mek1 kinase that prevents progression to the meiotic divisions by inhibiting the Ndt80 transcription factor [75,85]. In mammals, HORMAD2 is not required to promote DSB formation or synapsis, unlike HORMAD1, but is required to recruit the ATR kinase to unsynapsed regions on the sex chromosomes during male meiosis to implement the transcriptional silencing of these regions, which is needed for meiotic progression beyond pachytene [10,11]. HORMAD1 is also required for ATR recruitment to unsynapsed regions, but as HORMAD2 recruitment to axial elements is largely dependent on HORMAD1 [10], HORMAD1 may ensure ATR recruitment mostly by promoting HORMAD2 loading. In *C. elegans*, meiotic HORMADs are required for different quality control mechanisms that control meiotic progression. HTP-1 is required for the delayed exit from early prophase triggered by the presence of unsynapsed chromosomes, and to delay SC assembly until homology search is satisfied [15,16]. Similarly, defects in the formation of crossover precursors induce delayed exit from DSB-competent stages, in a process that requires HTP-1 and HTP-3 [86,87]. In worms, HORMAD signaling is thought to orchestrate meiotic progression mostly by regulating CHK-2 activity [73,88]. In addition to delaying early meiotic progression, the presence of unsynapsed chromosomes or unrepaired DSBs in pachytene nuclei triggers an apoptotic response via the activity of synapsis and DNA damage checkpoints [53]. The synapsis checkpoint requires PCH-2, HTP-1, HTP-3, and HIM-3 [53,89]. How HORMADs are capable of sensing different meiotic defects to generate signals that control meiotic progression and apoptosis is not well understood, but could involve changes in HORMA domain conformation, as well as the formation and dissociation of different protein complexes, as observed for MAD2 and REV7 and as suggested by recent studies in yeast [62]. Similar to the role of MAD2 in the spindle assembly checkpoint, soluble pools of HORMADs are also thought to be important components of meiotic checkpoints [62,90].

## 9. Regulation of Meiotic HORMADs by Post Translational Modifications

Phosphorylation events on Mad2 are known to regulate its conformational transition and to control its affinity for different ligands [91]. Similarly, recent studies show that phosphorylation events on meiotic HORMADs, and on the CMs that they bind, also regulate the function of these proteins. Phosphorylation of an [S/T]Q cluster on the C-terminus of yeast Hop1 by Mec1/Tel1 (ATM/ATR) promotes the recruitment and activation of the Mek1 kinase to axial elements, ensuring inter-homologue recombination and checkpoint activation [75]. The PP4 phosphatase also interacts with Hop1 and is proposed to promote the initial steps of Hop1 loading to axial elements [92]. In *C. elegans*, phosphorylation on the short N-terminal region preceding the HORMA domain of HTP-1 and HTP-2 promotes the recruitment of LAB-1 to axial elements to regulate sister chromatid cohesion release [19]; phosphorylation of S325 by MPK-1 on the C-terminus of HTP-1 regulates SC assembly and/or stability [93]; and phosphorylation of HIM-3 on the CM at its C-terminus regulates the binding of HTP-2 and SC disassembly [94]. In *Arabidopsis*, CDKA;1-dependent phosphorylation of residues within the HORMA domain promotes ASY1 loading to axial elements by increasing its binding affinity to ASY3 [57]. In addition to the above functional studies, phospho-specific antibodies, raised against S375 in a [S/T]Q motif of mouse HORMAD1, confirm in vivo phosphorylation of this residue [95]. Interestingly, HORMAD1 phosphorylation is detected in the absence of SPO11 activity, suggesting that HORMAD1 is phosphorylated in a DNA damage-independent manner [96]. Thus, phosphorylation events regulate meiotic HORMADs in at least three of the following ways: (1) by regulating the affinity of the HORMA domain for CMs; (2) by modifying the affinity of CMs for the HORMA domain; and (3) by regulating their ability to recruit interactors to their N- and C-terminal regions flanking the HORMA domain (Figure 5). Given the central role that kinases play in HORMAD-controlled events, including recombination and synapsis, it is likely that the examples discussed above constitute a small subset of the phosphorylation events that govern meiotic HORMADs functions.

In addition to phosphorylation, SUMOylation has also emerged as a regulator of meiotic HORMADs. For example, the RNF212 SUMO ligase is required to ensure that HORMAD1 reassociates with unsynapsed chromosomes during late meiotic prophase, suggesting that HORMAD1 behavior at this stage could be regulated by SUMOylation [77]. Recently, proteomic approaches have confirmed that both Hop1 and Red1, including its CM, are SUMOylated [97]; therefore, it is possible that both Hop1’s function and its recruitment to Red1’s CM may be controlled by SUMOylation. Clarifying how different functions of meiotic HORMADs are controlled by post translational modifications is a clear area for future research.

## 10. Conclusions

The orderly assembly and disassembly of protein complexes is an essential aspect of complex biological processes. Protein complexes containing HORMADs are at center stage of key meiotic events, including recombination, synapsis, and quality control mechanisms. Given the numerous interactors characterized for REV7 and MAD2, and the additional N- and C-terminal regions of meiotic HORMADs, it is likely that further uncharacterized interactors that enable meiotic HORMADs to partake in their diverse functions exist. Moreover, the presence of internal CMs on meiotic HORMADs endows these proteins with an added level of control over conformational changes that is not available to REV7 and MAD2. Understanding how conformational changes, interactor recruitment, and posttranslational modifications of meiotic HORMADs are integrated to promote recombination, synapsis, and quality control of meiosis remain important questions.

**Table 2 genes-13-00777-t002:** Comparison of protein interactors of well-characterized HORMAD proteins MAD2 and REV7 with meiotic HORMADs. * Proteins indicated have been suggested to interact via the safety belt-CM mechanism, based on sequence similarities and evidence of a direct interaction. ^‡^ Kd measurements have used truncated proteins, consisting of just their HORMA domain. ^†^ Proteins indicated have been characterized interacting outside of the HORMA domain. ^#^ No evidence for direct interaction. Immunoprecipitation and/or fluorescence colocalization dependency suggests an interaction. Background color means MAD2 and REV7 are not meiotic HORMADs.

.	Protein	Additional Structural Features	Interactors through Safety Belt-CM	Kd/μM	Interactors through an Alternate Interface	Interactors through an Uncharacterized Interface	References
*H. sapiens*	MAD2	-	MAD1	1.04	MAD2	-	[25,37,98,99]
			CDC20	0.1	p31^comet^		[26,98,100]
			SGO2	0.69	TRIP13		[32,101]
			REV3 *		BUBR1		[102,103]
			RIT1 *		WT1		[104,105]
*H. sapiens*	REV7	-	SHLD3	0.013 ± 0.0004	REV7	CLTA	[28,36,106,107]
			RAN	1.85	MAD2	HCCA2	[36,108,109]
			REV3		p31^comet^	PRCC	[35,36,110,111]
			IpaB		REV1	SIM2	[28,38,108,112]
			CAMP		SHLD2	TCF4	[28,39,113,114]
			ELK-1 *		TRIP13	ADP/E3-11.6K	[39,115,116]
			MDC9 *		CDH1	TF11-1	[102,117,118]
*S. cerevisiae*	Hop1	C-ter CM	Hop1 *	6.1 ± 1 ^‡^	Pch2	Mer2	[27,60,71]
		zinc finger	Red1	0.34 ± 0.03 ^‡^	Mek1 ^†^	PP4 ^#^	[27,92,119]
*S. pombe*	Hop1	C-ter CM	Rec10 *		Rec15 ^†^		[43]
		zinc finger					
*H. sapiens*	HORMAD1	C-ter CM	HORMAD1 *				[40]
			HORMAD2 *				[40]
			MCM9 *				[120]
*M. musculus*	HORMAD1	C-ter CM			TRIP13 ^#^	Cohesin ^#^	[10,33,45]
						IHO1	[70]
	HORMAD2	C-ter CM	HORMAD2 *	7.1 ± 0.5 ^‡^			[44]
			SYCP2 *				[44]
			HORMAD1 *				[10]
*C. elegans*	HTP-1	C-ter CM	HIM-3	0.7		LAB-1 ^#^	[19,40,88]
			HTP-3 motif #1	0.3			[40,88]
			HTP-3 motif #6 *	0.9			[40,88]
	HTP-2	C-ter CM	HIM-3	3.1			[40]
			HTP-3 motif #1	0.2			[40]
			HTP-3 motif #6	0.3			[40]
	HIM-3	C-ter CM	HTP-3 motif #4	0.3			[40,88]
			HTP-3 motifs #2, 3, 5 *				[40,88]
	HTP-3	6 C-ter CM				MRE-11/RAD-50 ^#^	[14]
						Cohesin ^#^	[40]
*A. thaliana*	ASY1	C-ter CM	ASY1 *		COMET		[44,57,58]
		SWIRM domain	ASY3 *		PCH2 ^#^		[41,44]
	ASY2						[121]
*Oryza sativa*	PAIR2	C-ter CM SWIRM domain	PAIR3 *		CRC1 (PCH2)		[56,122]

## Figures and Tables

**Figure 1 genes-13-00777-f001:**
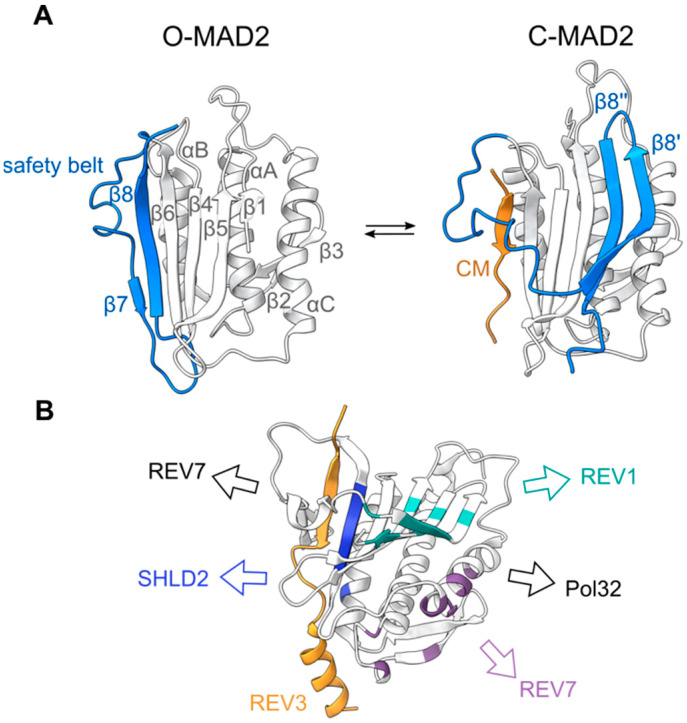
(**A**) Structures of human MAD2 in an open (PDB 1DUJ) and closed (PDB 1KLQ) conformation [24,26]. The safety belt and CM are highlighted in blue and orange, respectively. (**B**) In addition to the canonical safety belt-CM binding site (shown here by the REV3 CM; PDB 6BC8), REV7 interacts with other protein partners through additional interfaces. Important residues for these interactions between human REV7 and REV1 (cyan), REV7 (purple) and SHLD2 (blue) are highlighted. Additional interaction surfaces characterized in yeast Rev7 are shown in black [30].

**Figure 2 genes-13-00777-f002:**
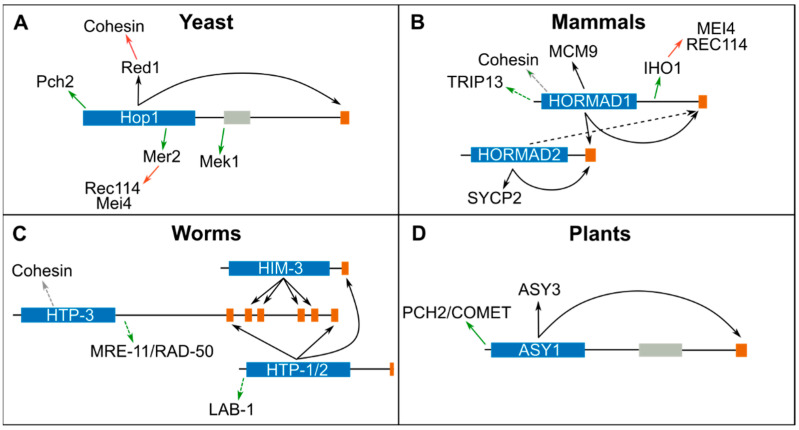
Summary of meiotic HORMAD interactors in yeast (**A**), mammals (**B**), worms (**C**) and plants (**D**). The HORMA domain of each protein is represented as a blue box and CMs by orange rectangles. Black arrows indicate interactions mediated through the safety belt-CM, green arrows indicate interactions through another interface, grey arrows indicate interactors through an unknown interface and red arrows indicate interactions not involving HORMADs. Interactions only supported by immunoprecipitation and/or fluorescence colocalization dependency are indicated with a dotted line (Table 2). See also Figure 3 for a representation of potential conformational changes in meiotic HORMADs.

**Figure 3 genes-13-00777-f003:**
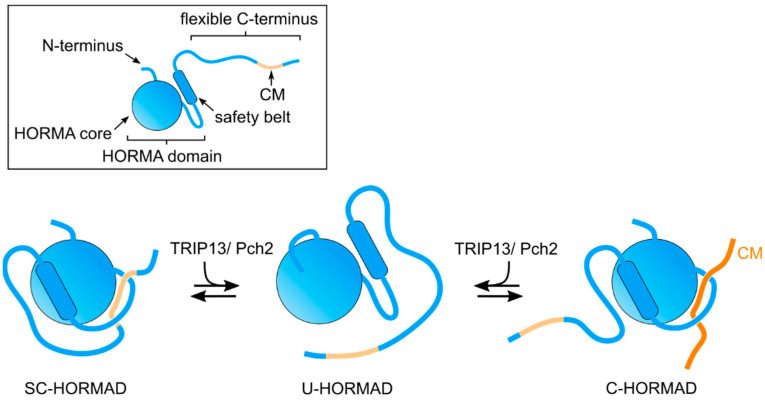
Graphical representation of possible meiotic HORMAD conformations, mediated by Pch2/TRIP13: SC (self-closed)-HORMAD; U(unbuckled)-HORMAD; and C(closed)-HORMAD bound to a CM motif on an interactor, such as an axis component. We note that, in addition to the HORMAD binding to its own CM in cis, a trans conformation is also possible.

**Figure 4 genes-13-00777-f004:**
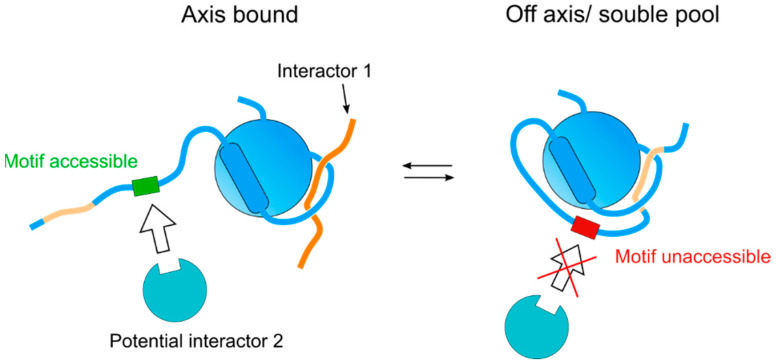
Graphical representation of a possible self-regulatory pathway utilized by meiotic HORMADs. Interactor 1 represents a canonical safety belt-CM interaction, interactor 2 represents a binding partner mediated through another interface. Although the additional hypothetical motif is represented here on the flexible C-terminus, it represents any motif within the protein that could be affected upon a conformation change.

**Figure 5 genes-13-00777-f005:**
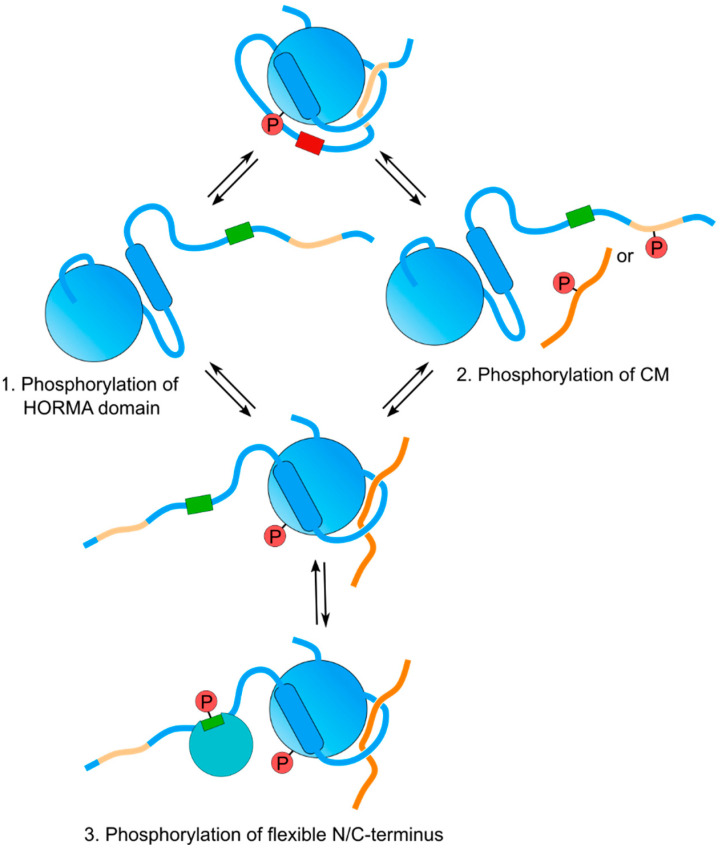
Modes of HORMAD regulation by phosphorylation. Though depicted separately, it is likely that multiple modes of regulation occur at the same time to finely tune meiotic HORMAD activity.

**Table 1 genes-13-00777-t001:** Orthologues of meiotic HORMADs, their main axis interactors, and their regulators in yeast, mammals, worms, and plants.

	*S. cerevisiae*	*Homo sapiens*/ *Mus musculus*	*C. elegans*	*Arabidopsis thaliana*
HORMADs	Hop1	HORMAD1	HTP-1	ASY1
		HORMAD2	HTP-2	ASY2
			HTP-3	
			HIM-3	
HORMAD interactor on axis	Red1	SYCP2	-	ASY3
		Cohesin	Cohesin	

HORMAD regulation	Pch2	TRIP13	PCH-2	PCH2
		p31^COMET^	CMT-1	COMET

## Data Availability

Not applicable.

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
