# Peer review of "Functions and Regulation of Meiotic HORMA-Domain Proteins"

_genes, 2022, doi:10.3390/genes13050777_

Round 1

Reviewer 1 Report

Overall, Prince and Martinez-Perez have written a very nice review about meiotic HORMA-domain proteins, focusing on genetic and structural regulatory components. As this is not my expertise, I enjoyed reading this review and learning about these domains. 

One very minor comment: the "safety belt" analogy is not necessary and I found it to be distracting. I completely understand where you are coming from and acknowledge that you use it throughout your review; however, I think your review would benefit from transitioning to describing these conformational states as "open" and "closed" (as opposed to "unbuckled" and "closed"). This is more consistent with the structural biology field. Likewise, your beta8'-beta8" region can be a "regulatory segment" as opposed to the "safety belt." 

Author Response

Reviewer 1:

Overall, Prince and Martinez-Perez have written a very nice review about meiotic HORMA-domain proteins, focusing on genetic and structural regulatory components. As this is not my expertise, I enjoyed reading this review and learning about these domains. 

We thank the reviewer for their overall support of our review.

One very minor comment: the "safety belt" analogy is not necessary and I found it to be distracting. I completely understand where you are coming from and acknowledge that you use it throughout your review; however, I think your review would benefit from transitioning to describing these conformational states as "open" and "closed" (as opposed to "unbuckled" and "closed"). This is more consistent with the structural biology field. Likewise, your beta8'-beta8" region can be a "regulatory segment" as opposed to the "safety belt." 

We have decided to keep the “safety belt” term as this is now broadly used in the literature of meiotic HORMADs. We use the terms open and closed to refer to the conformational states of Mad2 and Rev7, which are well characterised at the structural level, while the term “unbuckled” is used to refer to an open-like conformation present in meiotic HORMADs and that so far has been only partially characterised for yeast Hop1. We now indicate this clearly in the main text (lines 131-138). We have also removed the term unbuckled from the abstract (line 18) as this sentence refers to well characterised open and closed conformational states of Mad2 and Rev7.

Reviewer 2 Report

This is a comprehensive review that compares and contrasts the functions of HORMAD proteins in yeasts, mammals, worms, and plants. The review focuses on summarizing studies defining protein complex that form by HORMAD proteins, how meiotic HORMADs bind to the chromosome axis, the role of Pch2/TRIP3 in regulating their assembly and disassembly, the roles HORMADs play in recombination, SC assembly and checkpoint regulation, and finally how meiotic HORMADs are regulated by post-translational modifications. Overall, the article will be useful for those directly studying HORMADs and their interacting proteins since it does a good job of cataloging the current data.

1. Parts 2, 3, and 4 describe the binding properties of HORMADs. These sections are very dense and difficult to navigate for a non-expert, especially being at the start of the review. I might suggest inverting the review so that the sections on the roles of HORMADs and associated proteins and their regulation in meiosis are presented first. This would set the stage for the more detailed description of the binding features of these proteins. A few sentences at the end of the introduction to state the purpose/intent of the review and a roadmap of the various sections would help.

2. An additional table showing which HORMAD proteins across species are functional orthologs as well as the orthologs of their binding partners (e.g. TRIP13/Pch2) would help the reader navigate the article, especially the description of the proteins’ interacting partners in different species. With so many factors and with so many also having orthologs with different names it is really difficult to follow some sections.

3. I found part 4 especially difficult. I think it would help if you labeled the panels A, B, C, and D and then in the text refer to them directly. It would have also helped to see the cartoon version of HORMAD proteins before the interaction maps.

4. Given the placement of Table 1 near the end of article and separation from figure 2, it is difficult to go back and forth and read the text at the same time.

5. The call out to figure 2 on line 113 seems to refer to  figure 1

Minor point:
There are several instances (e.g. line 91 and 95) of the phrase “characterized interacting”. I suggest  “characterized as interacting” or writing it in another way e.g “have been shown to interact”

Author Response

This is a comprehensive review that compares and contrasts the functions of HORMAD proteins in yeasts, mammals, worms, and plants. The review focuses on summarizing studies defining protein complex that form by HORMAD proteins, how meiotic HORMADs bind to the chromosome axis, the role of Pch2/TRIP3 in regulating their assembly and disassembly, the roles HORMADs play in recombination, SC assembly and checkpoint regulation, and finally how meiotic HORMADs are regulated by post-translational modifications. Overall, the article will be useful for those directly studying HORMADs and their interacting proteins since it does a good job of cataloging the current data.

We thank the reviewer for their overall support of our review.

  1. Parts 2, 3, and 4 describe the binding properties of HORMADs. These sections are very dense and difficult to navigate for a non-expert, especially being at the start of the review. I might suggest inverting the review so that the sections on the roles of HORMADs and associated proteins and their regulation in meiosis are presented first. This would set the stage for the more detailed description of the binding features of these proteins. A few sentences at the end of the introduction to state the purpose/intent of the review and a roadmap of the various sections would help.

In our opinion, a description of meiotic HORMAD structure in relation to the better characterised HORMA proteins Mad2 and Rev7 prior to describing the specific roles of meiotic HORMADs, provides a logical framework to better understand their behaviour. However, we agree with the reviewer that providing a clear roadmap of the intent of the review at the end of the introduction will help the reader and therefore we have now incorporated this section at the end of the introduction. 

  1. An additional table showing which HORMAD proteins across species are functional orthologs as well as the orthologs of their binding partners (e.g. TRIP13/Pch2) would help the reader navigate the article, especially the description of the proteins’ interacting partners in different species. With so many factors and with so many also having orthologs with different names it is really difficult to follow some sections.

Following the reviewer’s suggestion, we have added a new table (Table 1) listing orthologs of meiotic HORMADs and their key interactors across species.

  1. I found part 4 especially difficult. I think it would help if you labeled the panels A, B, C, and D and then in the text refer to them directly. It would have also helped to see the cartoon version of HORMAD proteins before the interaction maps.

We have added the labeling to the panels of figure 2 and their corresponding references in the main text. We also now make an earlier reference on the main text and in the figure legend of figure 2 to the cartoon version of the HORMADs shown on figure 3.

  1. Given the placement of Table 1 near the end of article and separation from figure 2, it is difficult to go back and forth and read the text at the same time.

As suggested by the reviewer in point 2 above, we have added a table (new table 1) describing homology relationships of the main players mentioned in the review across yeast, mammals, worms, and plants. This table is now placed immediately before Figure 2 to facilitate comparisons. Due to its large size, it will be difficult to show old Table 1 (now Table 2) and Figure 2 on the same page, therefore we are keeping this table at the end of the manuscript as a summary of all interactions discussed in the review. If the editors see that it is feasible to move  Table 2 to an earlier location once it is on its final format and size, we are open to changing its placement.  

  1. The call out to figure 2 on line 113 seems to refer to  figure 1

The call out mentioned by the reviewer refers to the binding of the HORMA domain to a CM on the C-terminus of the same molecule. This type of CM binding is depicted on Figure 2A, but also perhaps more clearly on the cartoon of Figure 3. Therefore, we have also included a call out to Figure 3.  

Minor point:
There are several instances (e.g. line 91 and 95) of the phrase “characterized interacting”. I suggest  “characterized as interacting” or writing it in another way e.g “have been shown to interact”

We have made the corrections suggested by the reviewer.